# $A^\star$MCTS: Search with Theoretical Guarantee using Policy and Value Functions

## Abstract

Combined with policy and value neural networks, Monte Carlos Tree Search (MCTS) is a critical component of the recent success of AI agents in learning to play board games like Chess and Go (Silver et al., 2017). However, the theoretical foundations of MCTS with policy and value networks remains open. Inspired by MCTS, we propose $A^\star MCTS$, a novel search algorithm that uses both the policy and value predictors to guide search and enjoys theoretical guarantees. Specifically, assuming that value and policy networks give reasonably accurate signals of the values of each state and action, the sample complexity (number of calls to the value network) to estimate the value of the current state, as well as the optimal one-step action to take from the current state, can be bounded. We apply our theoretical framework to different models for the noise distribution of the policy and value network as well as the distribution of rewards, and show that for these general models, the sample complexity is polynomial in $D$, where $D$ is the depth of the search tree. Empirically, our method outperforms MCTS in these models.

## 1 Introduction

Monte Carlo Tree Search (MCTS) is an online heuristic search algorithm commonly used to find optimal policies for problems in reinforcement learning. It is a vital component of recent breakthroughs for AI in game play, including AlphaGo and AlphaZero (Silver et al., 2016; 2017).

An interesting difference between the MCTS used in AlphaGo/AlphaZero and traditional MCTS is the adoption of neural networks to predict the value of the current state (*value network*), and to prioritize the next action to search (*policy network*). The value network replaces random rollouts used to evaluate a non-terminal state, which saves both online computation time and substantially reduces the variance of the estimate. The policy network (coupled with PUCT (Rosin, 2011)) prioritizes promising actions for more fruitful exploration. Both networks are trained on pre-existing datasets (either from human replays or self-play) to incorporate prior knowledge. This strategy has been applied in other domains including Neural Architecture Search (Wang et al., 2018; Wistuba, 2017; Negrinho & Gordon, 2017; Wang et al., 2019b).

Despite the empirical success and recent popularity, theoretical foundations are lacking. (Kocsis & Szepesvári, 2006; Coquelin & Munos, 2007) analyze MCTS with rollouts using a multi-armed bandit framework and gives asymptotic results without finite sample complexity bounds. Powering MCTS with value/policy neural networks instead of rollouts is even more poorly understood.

To better understand how to efficiently learn optimal policies in decision trees using value and policy networks, we propose a novel algorithm, $A^\star MCTS$, which is a combination of $A^\star$ (Delling et al., 2009; Kanoulas et al., 2006) and MCTS. Like $A^\star$, it uses a priority queue to store all leaves currently being explored and picks the most optimistic one to expand, according to an upper confidence bound heuristic function. Like MCTS, it uses policy and value networks to prioritize the next state to be explored.

To facilitate the analysis, the policy and value networks are treated as black-box functions and a statistical model is built the accuracy of the predictions. In particular, we assume the standard deviation of the noisy estimates of intermediate state values decays (either polynomially or exponentially) as a function of the depth of the tree, so that the values of near-terminal states are estimated to a greater degree of accuracy. With this statistical model, we provide theoretical guarantees for the sample

complexity (expected number of state expansions / rollouts) to determine optimal actions. We apply our theoretical framework to simple models for rewards and show that our algorithm enjoys a sample complexity that is polynomial in depth $D$.

Our experiments validate the theoretical analysis and demonstrate the effectiveness of $A^\star$MCTS over benchmark MCTS algorithms with value and policy networks. To our knowledge, our work is the first that studies tree search for optimal actions in the presence of pre-trained value and policy networks, and we hope that it inspires further progress on this important problem.

## 2 Background and Related Work

Monte Carlo Tree Search (Browne et al., 2012) is an algorithm that tries to find optimal decisions in Markov Decision Processes by taking random samples in the decision space and building a tree based on the results. In each iteration, the agent traverses the tree by acting according to the tree policy (which usually involves some exploration) for the set of known states and then determines the value of an unexpanded state by performing random rollouts.

MCTS is commonly used to search for a one-step optimal actions from the root. The agent then takes this action, observes the result, transitions to a new state, and repeats MCTS starting from this new state, with the benefit of the additional observation. The main theoretical question of interest in MCTS is how quickly and efficiently one can find this optimal one-step action, in terms of the number of random rollouts, or the number of nodes in the tree that should be expanded. On the theoretical side, there is a small body of literature that addresses the theoretical foundations of MCTS with random rollouts (Kocsis & Szepesvári (2006), Veness et al. (2011)). The algorithm, called UCT (UCB applied to trees) models the search as a complex multi-armed bandit problem where the rewards are allowed to drift over time. The analysis shows that after a period of time $T$, only the optimal branch will be followed. Veness et al. (2011) gives various ways of improving UCT with variance reduction techniques.

Later followup work in Coquelin & Munos (2007) highlights some hard instances of UCT and offers some alternatives, including a modified UCT (with a different exploration bonus), Flat UCB, and BAST (smooth tree models).

On the experimental side, MCTS was popularized by the story of AlphaGo (Silver et al. (2016)), which used MCTS using a weighted combination of the value network's output and random rollouts to defeat the leading champion in Go. Follow-up work in AlphaZero (Silver et al. (2017)) used MCTS with only policy and value neural networks, without random rollouts and reports faster training with better performance. However, there is no theoretical foundation for the search algorithm that AlphaZero uses.

$A^\star$ (Nosrati et al., 2012) uses a priority queue to keep track of lowest cost actions. $A^\star$ combined with hand-designed heuristics, has been used in many path finding and routing problems (AlShawi et al., 2012; Rana & Zaveri, 2011; Ghaffari, 2014). Recent work (Wang et al., 2019a; Chen & Wei, 2011; Sigurdson & Bulitko, 2017) trains neural-based heuristic functions for $A^\star$ for specific problems. However, to the best of our knowledge, we are not aware of any prior work that performs theoretical analysis of such heuristic based $A^\star$ algorithms.

## 3 Preliminaries and Summary of Contributions

In this work we consider a deterministic, finite $D$-horizon Markov decision process (MDP) with no intermediate rewards. With the set of states denoted by $\mathcal{S}$, there is one deterministic root initial state $s_0$ and the process terminates at horizon $D$ with a termination reward that depends on the end state. We assume that there are exactly $K$ possible actions at every intermediate state $s$ and for simplicity that the process does not backtrack, i.e., each action leads to a new and unique state. This is a Markov decision tree.

The set of leaf states of the Markov decision tree are denoted as $\mathcal{L}$. We shall denote an arbitrary state at depth $d$ as $s_d$ and also a leaf state as $l \in \mathcal{L}$. There is a value function $V : \mathcal{S} \to \mathbb{R}$ that takes as input a state $s$ and returns the value of $s$, defined in the following way. If $s \in \mathcal{L}$, then $V_s$ is the termination reward of that particular leaf. For an intermediate node $s$, $V_s = \max_{l \in \mathcal{L}(s)} V_l$, where

| Summary of Notations Used | |
|---|---|
| $K$ | Branching factor |
| $V_{s_d}$ | True value of state $s_d$ at depth $d$ |
| $\Delta_{s_d}$ | Gap to optimal value ($V^* - V_{s_d}$) |
| $U_{s_d}$ | Predicted value of state $s_d$ by value network (deterministic) |
| $X_d$ | 0-mean, standard deviation $\sigma_d$ random variable for the noise of the value prediction $U_{s_d}$ at depth $d$. $V_{s_d} - U_{s_d} \sim X_d$. |
| $\sigma_d$ | standard deviation of $X_d$, fixed and known as part of the problem model. We assume that $\sigma_d$ decays with depth. |
| $c_d$ | exploration bonus, determined based on $\sigma_d$. $|X_d| \leq c_d$ with high probability |
| $U_{s_d} + c_d$ | priority value (value used to determine the ordering of which to nodes expand in the priority queue). |
| $U_{r_i}^{\pi}$ | Predicted value of child $i$ at some depth $d$ by policy network (deterministic) |
| $X_d^{\pi}$ | 0-mean, standard deviation $\sigma_d$ random variable for the noise of the value prediction of child $i$ at depth $d$, $U_{r_i}^{\pi}$. $V_{r_i} - U_{r_i}^{\pi} \sim X_d^{\pi}$. |

Table 1: Summary of main notations used in the paper.

$\mathcal{L}(s)$ is the set of leaf nodes that share $s$ as an ancestor, i.e., the set of leaf nodes in the subtree rooted at $s$. We assume that the highest value $V^* \equiv \max_{l \in \mathcal{L}} V_l = \max_{s \in \mathcal{S}} V_s$ is achieved at a unique leaf node $l^* = \arg\max_{l \in \mathcal{L}} V_l$.

In this work, we focus on the problem of efficiently computing $V^*$ exactly or approximately with a controlled accuracy without direct access to the value function $V : \mathcal{S} \to \mathbb{R}$, as a stepping stone to finding the optimal one-step action from $s_0$. Instead, there is a pre-trained black-box resource, *value network* $U : \mathcal{S} \to \mathbb{R}$ that takes as input a state and returns a noisy value estimate, i.e., for a state $s_d$ at depth $d$, $U_{s_d} \equiv V_{s_d} + X_{s_d}$, where $X_{s_d}$ are i.i.d. copies of a random variable $X_d$ that depends only on the depth $d$. We assume that at depth $D$, which is the level with all the leaf nodes, $U_l = V_l$ (i.e., the value network returns the ground truth values without any noise). The efficiency is defined in terms of sample complexity, which is the number of queries made to the value network for the value of a state. In the worst case, all leaves need to be queried and the sample complexity is $K^D$.

Another type of available pre-trained black box resource is a *policy network*. Let $\mathcal{C}(s) = \{r_1, \ldots r_K\}$ denote the set of $K$ children of state $s$, each corresponding to the transition state of one of the $K$ actions $\{a_1, \ldots a_K\}$. The policy network outputs a probability estimate of the form $p_{r_i} = \frac{e^{U_{r_i}^{\pi}}}{\sum_{k=1}^{K} e^{U_{r_k}^{\pi}}}$, where each $U_{r_i}^{\pi}$ is a perturbed estimate of $V_{r_i}$. Here the superscript $\pi$ indicates that this is the approximation from the policy network. We assume that if the child $r_i$ is at depth $d$ then the estimate $U_{r_i}^{\pi} = V_{r_i} + X_{r_i}^{\pi}$, where $X_{r_i}^{\pi}$ are i.i.d. copies of a random variable $X_d^{\pi}$ that depends only on the depth $d$. We *assume* that the distribution of $X_d$ and $X_d^{\pi}$ are known a priori for each depth $d$. Further, the variances of $X_d$ and $X_d^{\pi}$ decrease sufficiently rapidly as the depth $d$ increases (i.e., as one traverses deeper in the tree, the noise level is smaller). This assumption is motivated by basic approximate dynamic programming principles, which is that near-terminal states require a smaller sample complexity to estimate well, since the backwards induction will propagate this noise to earlier states, making it harder to estimate the values of initial states well.

### 3.1 MAIN CONTRIBUTIONS

We develop and analyze (giving specific sample complexity bounds) algorithms to estimate $V^*$ exactly and approximately using value and policy networks. Since our value and policy networks are trained functions that give fixed, deterministic noisy outputs for each state (no matter how many times one calls the function on a particular state), where the noise variance is smaller at deeper levels of the search tree, we use our value and policy network estimates as signals for which are the promising states to expand. We rank the states based on our optimistic estimates $U_{s_d} + c_d$ in a priority queue, and we expand (query the value network for the value estimate) the children of the top nodes of the priority queue to get increasingly more accurate value estimates and determine the value of the optimal leaf $V^*$.

The main contributions are organized in the following sections:

- Section 4 introduces and analyzes $A^\star$MCTS-$V^*$ and $A^\star$MCTS-$\pi^*V^*$ that are our core techniques for estimating $V^*$ exactly. $A^\star$MCTS-$V^*$ uses only a value network to determine $V^*$ exactly and outputs the optimal one-step from $s_0$ that leads to $l^*$. We give an expected sample complexity in Theorem 1. $A^\star$MCTS-$\pi^*V^*$ builds on $A^\star$MCTS-$V^*$ by using a policy network to further reduce the sample complexity, which we bound in Theorem 2.

- Section 5 extends $A^\star$MCTS-$V^*$ and $A^\star$MCTS-$\pi^*V^*$ to the case when we tolerate $\delta$-additive approximations of $V^*$.

- Section 6 introduces some sample distribution models for the value functions and the noise and applies our results to those models. We show that for some reasonable models, our algorithms can achieve sample complexity that is polynomial in depth $D$. We also provide some experimental comparisons against benchmark MCTS implementations.

## 4 FINDING $V^*$ EXACTLY USING VALUE AND POLICY NETWORKS

This section presents our main techniques, $A^\star$MCTS-$V^*$ and $A^\star$MCTS-$\pi^*V^*$, which find $V^*$ exactly using a value network only in the former and a value network and a policy network in the latter.

$A^\star$MCTS-$V^*$, Algorithm 1, uses only a value network to determine $V^*$ exactly and outputs the optimal one-step action from $s_0$ that takes the agent to an ancestor of $l^*$. The idea behind the algorithm is very simple and similar to $A^\star$ search. We first expand (i.e., query the value network) the children of the root node $s_0$, compute their value network estimate plus the exploration bonus, and then insert these nodes to a priority queue $Q$. The exploration bonus, denoted by $c_d$, is chosen such that $\mathbb{P}(|V_{s_d} - U_{s_d}| \leq c_d) \geq 1 - \frac{\beta}{DK^d}$. In each subsequent iteration, the algorithm picks the top element in the priority queue $Q$, which is the most optimistic state, and expands all the children of that state, and adds those children to $Q$. We terminate at the first time when the most optimistic element is a leaf node. Since the value network gives the values of leaf nodes exactly, this leaf node must be $l^*$.

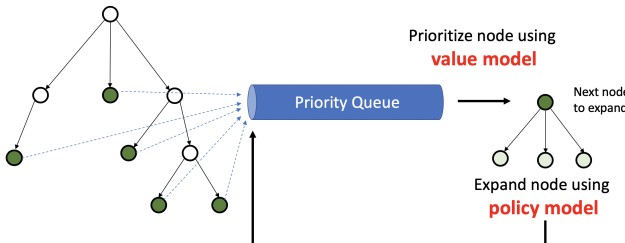

Figure 1: Illustration of our overall method. We prioritize nodes to expand using the value network estimates. Algorithm 1 queries the value network for all children of the first node in the priority queue. In our next algorithm, Algorithm 2, we show how to use a policy network to choose which children nodes to add to the priority queue.

We present the complexity bound in Theorem 1. The complexity bound is based on one very simple observation, which is that we end up choosing a sub-optimal node if the value network estimate of that node plus the upper bound on the possible error (optimism) is higher than the true optimal value. Otherwise we will not choose to expand that node, since we are sure that it will not be the optimal node. Therefore a sub-optimal internal node $s_d$ is chosen if $V^* \leq U_{s_d} + c_d = V_{s_d} + X_{s_d} + c_d$, where $U_{s_d}$ is the value network estimate for state $s_d$, and $c_d$ is the upper bound on the possible error for the value estimate. We should also account for the ancestors of $s_d$, if we are lucky enough to be able to rule out one of the ancestors of $s_d$, we would never expand beyond that ancestor to reach $s_d$, and $s_d$ would not be in our priority queue. Theorem 1 states this complexity bound.

**Theorem 1.** *With probability $1 - \beta$, Algorithm 1, $A^\star$MCTS-$V^*$, returns $V^*$. The expected sample complexity (number of calls to the value network) is*

$$\mathbb{E}[N] = KD + \sum_{s_d \in \mathcal{S}, s_d \notin \mathcal{L}, s_d \notin \mathcal{A}(l^*)} K \cdot \mathbb{P}\left( \Delta_{s_d} - X_{s_d} - c_d \leq 0 \bigcap_{s_{d'} \in \mathcal{A}(s_d)} \Delta_{s_{d'}} - X_{s_{d'}} - c_{d'} \leq 0 \right),$$

---

**Algorithm 1** $A^\star$MCTS-$V^*$

---

**Require:** Value Network with error distribution for $X_d$ at each depth $d \leq D$
 1: For each $d < D$, compute $c_d$ so that $\mathbb{P}(|X_d| \leq c_d) \geq 1 - \frac{\beta}{DK^d}$.
 2: $Q \leftarrow (s_0, U_{s_0} + c_d)$
 3: **while** the depth of $Q.front()$ is $< D$ **do**
 4:     $(s, \_) = Q.pop()$
 5:     **for** $r \in \mathcal{C}(s)$ **do**
 6:         $d \leftarrow$ depth of $r$
 7:         $Q.enqueue(r, U_r + c_d)$
 8:     **end for**
 9: **end while**
10: $(s, \_) \leftarrow Q.front()$
11: **Return** $U_s$ and the action at $s_0$ that leads to $s$

---

where $\mathcal{A}(l^*)$ is the set of ancestor states of the optimal leaf, $l^*$, and $\Delta_{s_d} = V^* - V_{s_d}$.

Before we give the proof, we briefly note that getting an exact expression for the expected sample complexity depends on the problem model (the distribution for the gaps $\Delta_{s_d}$ and the errors $X_{s_d}$). We give two examples in Section 6 of different models and we apply this theorem to those models to derive the sample complexity, which we show is polynomial in depth $D$. Please check Appendix for the proof.

### 4.1 COMBINING VALUE NETWORK WITH POLICY NETWORK

Suppose that in addition to the value network $U$, one has access to a policy network $U^\pi$ that outputs, for each child of $s$, $r_1, \ldots r_K$ (each corresponding to the transition state of a different action $a_1, \ldots a_K$) a probability estimate of the form:

$$p_{r_i} = \frac{e^{U^\pi_{r_i}}}{\sum_{k=1}^K e^{U^\pi_{r_k}}} \ ,$$

where each $U^\pi_{r_i} = V_{r_i} + X^\pi_{r_i}$ is a perturbed estimate of $V_{r_i}$. If $r_i$ is at depth $d$, the noise $X^\pi_{r_i}$ is an i.i.d. copy of the random variable $X^\pi_d$ that depends only on the depth $d$. Since all the probabilities are known, we assume without loss of generality that states $r_1, \ldots r_K$ are ordered so that $U^\pi_{r_1} \geq U^\pi_{r_2} \geq U^\pi_{r_3} \ldots \geq U^\pi_{r_K}$, or equivalently $p_{r_1} \geq p_{r_2} \geq p_{r_3} \ldots \geq p_{r_K}$.

The following algorithm $A^\star$MCTS-$\pi^*V^*$ uses both the value and the policy networks to determine $V^*$ exactly. When we expand a given state $s_d$, instead of querying for the value estimate for every child of $s_d$ and adding each child to the priority queue, we add the top 2 children (ordered by the probabilities given by the policy network) to the queue by default. For the $k$-th child node, where $k > 2$, we add this child if there is a small gap between the policy network probability of the $k-1$-th child and the probability of the top child, where the threshold for the gap is given by the noise model. Intuitively, we are saying that if this gap is sufficiently big, even after accounting for the noise in the policy network, there is no way that this $k$-th child is part of the optimal policy, so we do not have to add it to the queue.

The key insight behind this algorithm is that, if there is a large gap between the probabilities of the children as given by the policy network, one can infer that the children states associated with the smaller probabilities are not worth expanding because they have small values. This is made rigorous in the statement and proof of Theorem 2.

**Theorem 2.** *With probability $1 - \beta$, Algorithm 2, $A^\star$MCTS-$\pi^*V^*$, returns $V^*$. The expected sample complexity (number of calls to the value network) is*

$$\mathbb{E}[N] \leq \sum_{s_d \notin \mathcal{L}} \left( 2 + \sum_{k=2}^{K-1} \mathbb{P}\left( V_{r_1} - V_{r_k} + X^\pi_{(d+1),1} - X^\pi_{(d+1),k} \leq 2c^\pi_{d+1} \right) \right) \cdot$$

$$\mathbb{P}\left( \Delta_{s_d} - X_{s_d} - c_d \leq 0 \bigcap_{s_{d'} \in \mathcal{A}(s_d)} \Delta_{s_{d'}} - X_{s_{d'}} - c_{d'} \leq 0 \right),$$

---

**Algorithm 2** $A^\star$MCTS-$\pi^*V^*$

---

**Require:** Value Network, Policy Network, failure tolerance $\beta$
 1: For each $d < D$, compute $c_d$ so that $\mathbb{P}(|X_d| \leq c_d) \geq 1 - \frac{\beta}{2DK^d}$.
 2: For each $d < D$, compute $c_d^\pi$ so that $\mathbb{P}(|X_d^\pi| \leq c_d^\pi) \geq 1 - \frac{\beta}{2DK^d}$.
 3: $Q \leftarrow (s_0, U_{s_0})$
 4: **while** the depth of $Q.front()$ is $< D$ **do**
 5:     $(s, \_) = Q.pop()$
 6:     **for** $i \in [1, \ldots, K]$, ordered according to the probabilities of the policy network **do**
 7:         $d \leftarrow$ depth of $r_i$
 8:         **if** $i < 3$ or $\ln\left(\frac{p_{r_1}}{p_{r_{i-1}}}\right) \leq 2c_d^\pi$ **then**
 9:             $Q.enqueue(r_i, U_{r_i} + c_d)$
10:         **else**
11:             break;
12:         **end if**
13:     **end for**
14: **end while**
15: $(s, \_) \leftarrow Q.front()$
16: **Return** $U_s$ and the action at $s_0$ that leads to $s$

---

where $\Delta_{s_d} = V^* - V_{s_d}$, and $X^\pi_{(d+1),k}$ *is the random variable associated with $r_k$ for the policy network, i.e., $U^\pi_{r_k} = V_{r_k} + X^\pi_{(d+1),k}$.*

As with Theorem 1, getting an exact expression for the expected sample complexity depends on the problem model (the distribution for the gaps $\Delta_{s_d}$ and the errors $X_{s_d}$). We give two examples in Section 6 of different models and we apply this theorem to those models to derive the sample complexity, which we show is polynomial in depth $D$. Please check Appendix for the proof.

## 5 APPROXIMATING $V^*$ USING VALUE AND POLICY NETWORKS

This section extends the algorithms to the problem setting where we are willing to tolerate an additive error $\delta$ in the estimate for $V^*$

We first consider the extension of $A^\star$MCTS-$V^*$, Algorithm 1. The key idea is that it suffices to stop searching at a depth $\widetilde{D}$ whenever $c_{\widetilde{D}} \leq \delta$. As long as the most optimistic state at depth $\widetilde{D}$ is identified, the value of this state approximates $V^*$ up to an additive error $\delta$. The main result is stated in Theorem 3 and the algorithm is Algorithm $A^\star$MCTS-$\hat{V}$, presented in Appendix 8.3.

**Theorem 3.** *With probability $1 - \beta$, Algorithm 3, $A^\star$MCTS-$\hat{V}$, returns $V^*$ up to additive error $\delta$. The expected sample complexity (number of calls to the value network) is*

$$\mathbb{E}[N] \leq K\widetilde{D} + \sum_{\substack{s_d \in \mathcal{S} \\ d < \widetilde{D} \\ s_d \notin \mathcal{A}(l^*)}} K \cdot \mathbb{P}\left(\Delta_{s_d} - X_{s_d} - c_d - c_{\widetilde{D}} \leq 0 \bigcap_{s_{d'} \in \mathcal{A}(s_d)} \Delta_{s_{d'}} - X_{s_{d'}} - c_{d'} - c_{\widetilde{D}} \leq 0\right)$$

where $\mathcal{A}(l^*)$ *is the set of ancestor states of the optimal leaf, $l^*$, and $\Delta_{s_d} = V^* - V_{s_d}$, and $\widetilde{D}$ is the minimum depth at which $c_{\widetilde{D}} \leq \delta$*

Using the same reasoning as the one for $A^\star$MCTS-$\hat{V}$, we can extend $A^\star$MCTS-$\pi^*V^*$ to the following result.

**Theorem 4.** *With probability $1 - \beta$, one can find $V^*$ up to additive approximation error $\delta$ using expected sample complexity (number of calls to the value network):*

$$\mathbb{E}[N] \leq \sum_{s_d, d < \widetilde{D}} \left( 2 + \sum_{k=2}^{K-1} \mathbb{P}\left( V_{r_1} - V_{r_k} + X^{\pi}_{(d+1),1} - X^{\pi}_{(d+1),k} \leq 2c^{\pi}_{d+1} \right) \right) \cdot$$

$$\mathbb{P}\left( \Delta_{s_d} - X_{s_d} - c_d - c_{\widetilde{D}} \leq 0 \bigcap_{s_{d'} \in \mathcal{A}(s_d)} \Delta_{s_{d'}} - X_{s_{d'}} - c_{d'} - c_{\widetilde{D}} \leq 0 \right) ,$$

*where $\widetilde{D}$ is the minimum depth at which $c_{\widetilde{D}} \geq \delta$.*

## 6 MODELS AND EXPERIMENTS

In this section, we discuss two simple yet representative models for the Markov decision tree and show that the sample complexity is polynomially bounded under reasonable noise models.

### 6.1 CHOOSING $c_d$ FOR GAUSSIAN NOISE

The noises $X_d$, $X^{\pi}_d$ are taken to be Gaussian $\mathcal{N}(0, \sigma_d^2)$ random variables. Two noise models are tested. The first is a Gaussian with an exponentially decaying standard deviation, i.e., $X_d, X^{\pi}_d \sim \mathcal{N}(0, \frac{1}{\alpha^{2d}})$ for some $\alpha > 1$. The second is a Gaussian with a polynomially decaying standard deviation, i.e., $X_d, X^{\pi}_d \sim \mathcal{N}(0, \frac{1}{d^{2\gamma}})$, for some $\gamma > 1$. The experiments are carried out for $\alpha = 1.3$ and $1.5$ for the exponential case and $\gamma = 1.3$ and $1.5$ for the polynomial case.

The assumption of decaying standard deviation as a function of depth is motivated by basic approximate dynamic programming principles, which we elaborate on in Section 3.

The choice of $c_d$ is essential to the performance of the algorithms. Standard concentration inequalities for sub-Gaussian random variables state that $\mathbb{P}(|X_d| \geq c_d) \leq 2e^{\frac{-c_d^2}{2\sigma_d^2}}$, and this probability expression is required to be less than $\frac{\beta}{DK^d}$. After substituting it in, we arrive at $c_d = O(\sqrt{d}\sigma_d)$.

### 6.2 CONSTANT GAP MODEL

In the constant gap model, $V^* - V_{l^*} = \eta$ for a fixed constant $\eta > 0$ and $V_l = 0$ for any other leaf $l \neq l^*$. Therefore, the gap $\Delta(l) = \eta$ for any $l \neq l^*$. The noise variables $X_d, X^{\pi}_d$ are mean-zero Gaussian random variables.

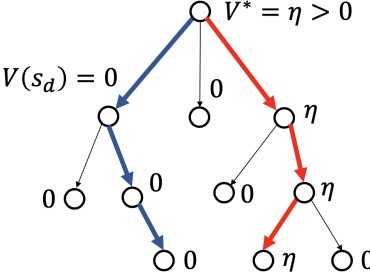

Figure 2: The constant gap model. The leaves have value $0$ except for the optimal leaf, which has value $\eta$. A blue path is a sub-optimal path, the red path is the optimal path.

**Using value network.** The expected sample complexity for Algorithm 1 is

$$\mathbb{E}[N] = KD + \sum_{d=1}^{D-1} \sum_{i=0}^{d-1} \left[ (K-1) \prod_{j=0}^{i} K\mathbb{P}(\eta - X_{(d-j)} - c_{(d-j)} \leq 0) \right] .$$

This translates to a sample complexity of $\mathbb{E}[N] \leq KD + D^2(K-1)K^c$, for the smallest constant $c$ that satisfies $\frac{\eta}{\sigma_c} - \sqrt{c} \geq \sqrt{2 \log K}$.

The analysis for this result, as well as similar analyses and sample complexity results for Algorithm 2 and the approximate counterparts are in Appendix 8.4.

**Experiments.** We compare the performance of $A^\star\text{MCTS-}V^*$ with the MCTS algorithm that uses the UCB estimate $Q_{s,a} + 2c\sqrt{\frac{\log(\sum_b N_{s,b})}{N_{s,a}}}$ and queries the value network $U_s$ instead of expanding state $s$ with random rollouts, where $c$ is a constant, and $N_{s,b}$ are visitation counts for choosing action $a$ from state $s$. We also compare $A^\star\text{MCTS-}\pi^*V^*$ with the PUCT algorithm described in Silver et al. (2017) that uses the UCB estimate $Q_{s,a} + c \cdot p_{s,a} \cdot \sqrt{\frac{\sum_b N_{s,b}}{N_{s,a}}}$, where $p_{s,a}$ is the probability given by the policy network $U^\pi$.

The experiments are performed on a tree of depth $D = 10$ with $K = 5$ children per state. As defined in the constant gap model, the optimal leaf has reward $\eta$ and all other leaves have reward $0$, and we experiment with $\eta = 1$ and $\eta = 0.5$. In our experiments, we use $c = 1$ in the the bonus UCB expression for benchmark MCTS and PUCT algorithms. Our algorithms set $c_d = 5 \cdot \sqrt{d}\sigma_d$, where $\sigma_d$ is the variance of the Gaussian noise. This choice is justified earlier in Section 6.1.

Since the main use case for these algorithms is to find the optimal one-step action from $s_0$, we collect data on the one-step action that the algorithm would output at every $1,000$ expansions (or queries to the value network $U$). MCTS algorithms would output the one-step action from $s_0$ with the highest visitation count. Our algorithms would output the action from $s_0$ that leads to $s$, where $s$ is the highest-valued element in the current priority queue $Q$ based on the value network estimate $U_s$. We give an overall budget of 20,000 expansions and note whether or not (so the data is 1 or 0) the algorithm chooses the right one-step action. Each of our experiments average over 200 trials. Tables 1 and 2 give the proportion of trials that return the correct optimal one-step action at the 20,000-th expansion. To give an idea about the performance gain, Figures 3 and 4 show the success proportion over time, at successive 1000 expansion intervals for Algorithm 2 and PUCT for $\eta = 0.5$ in both noise models (polynomially and exponentially decaying).

| | Polynomially Decaying Noise | | | | Exponentially Decaying Noise | | | |
|---|---|---|---|---|---|---|---|---|
| | $\gamma = 1.3$ | | $\gamma = 1.5$ | | $\alpha = 1.3$ | | $\alpha = 1.5$ | |
| | Alg 1 | MCTS | Alg 1 | MCTS | Alg 1 | MCTS | Alg 1 | MCTS |
| $\eta = 1$ | **1** | 0.51 | **1** | 0.695 | **1** | 0.355 | **1** | 0.605 |
| $\eta = 0.5$ | **1** | 0.38 | **1** | 0.435 | **0.65** | 0.265 | **1** | 0.4 |

Table 2: Value Network Only Results for the Constant Gap Model

| | Polynomially Decaying Noise | | | | Exponentially Decaying Noise | | | |
|---|---|---|---|---|---|---|---|---|
| | $\gamma = 1.3$ | | $\gamma = 1.5$ | | $\alpha = 1.3$ | | $\alpha = 1.5$ | |
| | Alg 2 | PUCT | Alg 2 | PUCT | Alg 2 | PUCT | Alg 2 | PUCT |
| $\eta = 1$ | 1 | 1 | 1 | 1 | 1 | 1 | 1 | 1 |
| $\eta = 0.5$ | **1** | 0.885 | **1** | 0.92 | 0.685 | **.705** | **1** | 0.875 |

Table 3: Value and Policy Network Results for the Constant Gap Model

## 6.3 GENERATIVE MODEL

The model generates the value function of the nodes in the Markov decision tree in a recursive way. This model first chooses the optimal value $V_{s_0} = V^*$ at the root $s_0$. Given the value function of a parent node $p$, the value functions for the children $r_1, \ldots, r_K$ are generated as follows: one of the children (without loss of generality, say $r_1$) has the same value function as the parent $V_p$; each of the remaining children $r_i$ has a value function $V_p - Y$ with $Y$ sampled independently from $\sim U[0, \eta]$, the uniform distribution over the interval $[0, \eta]$. At level $d$, there are exactly $\binom{d}{t}(K-1)^t$ states with value function distributed according to $V^* - (Y_1 + \ldots + Y_t)$. When $t$ is non-negligible, $Y_1 + \ldots + Y_t$ concentrates at constant value $t\eta/2$ by the concentration of measure phenomenon.

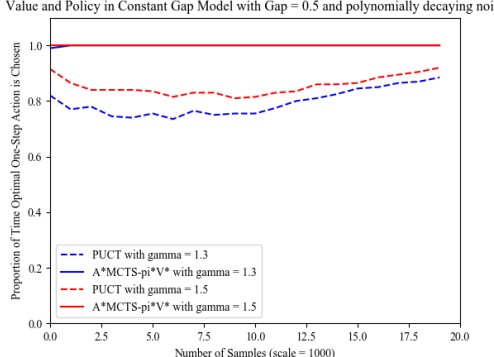

Figure 3: Using Value and Policy Networks for search in the Constant Gap Model with polynomially decaying noise and $\eta = 0.5$.

Figure 4: Using Value and Policy Networks for search in the Constant Gap Model with exponentially decaying noise and $\eta = 0.5$.

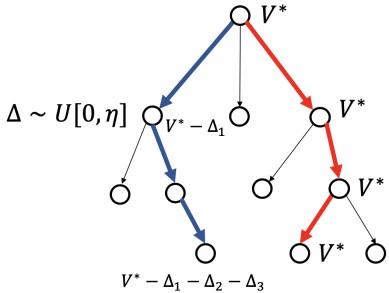

Figure 5: The generative model. A blue path is a sub-optimal path, the red path is the optimal path. The optimal leaf has value $V^*$. Sub-optimal nodes inherit the gaps of their parent as well as an extra gap $Y$, so that the sub-optimal leaf values get smaller and smaller.

**Using value networks.** Following the estimate in Theorem 1, the overall complexity for Algorithm 1 can be bounded up to an extra $KD$ term by

$$\sum_{d=1}^{D}\sum_{t=0}^{d} \binom{d}{t}(K-1)^t \mathbb{P}\left(X_d \geq \frac{\eta t}{2} - c_d\right) = \sum_{d=1}^{D}\sum_{t=0}^{d} \binom{d}{t}(K-1)^t \mathbb{P}\left(Z \geq \frac{\eta t}{2\sigma_d} - \widetilde{C}\sqrt{d}\right)$$

where $Z$ is a standard normal distribution. The goal is to bound this by a quantity polynomial (rather than exponential) in $D$ when $\eta$, $K$, and $\alpha$ are considered fixed. To proceed, we introduce $T(d) = \frac{2(\sqrt{2\log K}+1)\sqrt{d}\sigma_d}{\eta}$. For any $t \geq T(d)$,

$$\mathbb{P}\left(Z \geq \frac{\eta t}{2\sigma_d} - \widetilde{C}\sqrt{d}\right) \lesssim e^{-\log Kd} = K^{-d}.$$

Therefore, for a fixed $d$,

$$\sum_{t=T(d)}^{d} \binom{d}{t}(K-1)^t \mathbb{P}\left(Z \geq \frac{\eta t}{2\sigma_d} - \widetilde{C}\sqrt{d}\right) \leq \sum_{t=T(d)}^{d} \binom{d}{t}(K-1)^t K^{-d} \leq \sum_{t=0}^{d} \binom{d}{t}(K-1)^t K^{-d} = 1.$$

Therefore, up to a term linear in $D$, the sample complexity can be bounded by

$$\sum_{d=1}^{D}\sum_{t=1}^{T(d)} \binom{d}{t}(K-1)^t \mathbb{P}\left(Z \geq \frac{\eta t}{2\sigma_d} - \widetilde{C}\sqrt{d}\right). \tag{1}$$

Notice first that $\mathbb{P}(Z \geq \frac{\eta t}{2\sigma_d} - \sqrt{d})$ is always bounded by $1$. Second, since in both the exponential and polynomial noise models $\sigma_d$ decays faster than $1/\sqrt{d}$ asymptotically, $T(d) = \frac{2(\sqrt{2\log K}+1)\sqrt{d}\sigma_d}{\eta}$

is uniformly bounded by a constant $C(K, \eta, \alpha)$ due to the exponential decay in $d$. Therefore, the term (4) can be bounded by $(DK)^{C(K,\eta,\alpha)}$, which is polynomial (rather than exponential) in the depth $D$.

Similar analysis and sample complexity statements can be made for the performance of Algorithm 2 and the approximate counterparts. We defer this treatment to Appendix 8.5.

**Experiments.** Our experimental results for the Generative Model follow the same setup and parameters as the experiments for the Constant Gap Model in Section 6.2. Tables 3 and 4 give the proportion of trials that return the correct optimal one-step action at the 20,000-th expansion. Figures 6 and 7 show the success proportion over time, at successive 1000 expansion intervals for Algorithm 2 and PUCT for $\eta = 1$ in both noise models (polynomially and exponentially decaying).

| | | Polynomially Decaying Noise | | | Exponentially Decaying Noise | | | |
|---|---|---|---|---|---|---|---|---|
| | | $\gamma = 1.3$ | | $\gamma = 1.5$ | | $\alpha = 1.3$ | | $\alpha = 1.5$ |
| | Alg 1 | MCTS | Alg 1 | MCTS | Alg 1 | MCTS | Alg 1 | MCTS |
| $\eta = 1$ | **1** | 0.895 | **1** | 0.895 | **1** | 0.9 | **1** | 0.895 |
| $\eta = 0.5$ | **1** | 0.865 | **1** | 0.865 | 0.825 | **0.865** | **0.995** | 0.87 |

Table 4: Value Network Only Results for the Generative Model

| | | Polynomially Decaying Noise | | | Exponentially Decaying Noise | | | |
|---|---|---|---|---|---|---|---|---|
| | | $\gamma = 1.3$ | | $\gamma = 1.5$ | | $\alpha = 1.3$ | | $\alpha = 1.5$ |
| | Alg 2 | PUCT | Alg 2 | PUCT | Alg 2 | PUCT | Alg 2 | PUCT |
| $\eta = 1$ | **1** | 0.94 | **1** | 0.94 | **0.99** | 0.92 | **1** | 0.935 |
| $\eta = 0.5$ | **1** | 0.935 | **1** | 0.92 | 0.82 | **0.9** | **1** | 0.92 |

Table 5: Value and Policy Network Results for the Generative Model

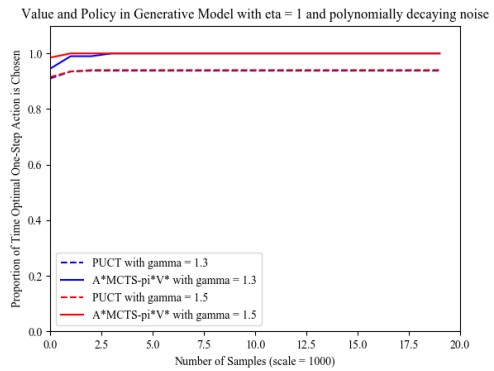

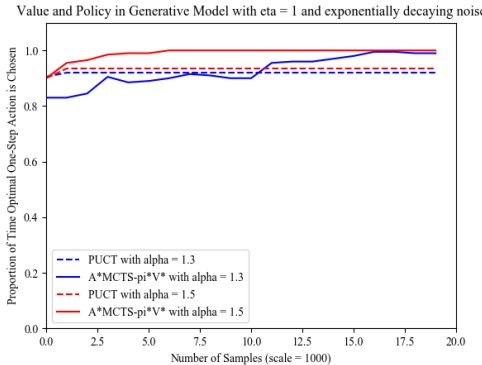

Figure 6: Using Value and Policy Networks for search in the Generative Model with polynomially decaying noise and $\eta = 0.5$.

Figure 7: Using Value and Policy Networks for search in the Generative Model with exponentially decaying noise and $\eta = 0.5$.

# 7 CONCLUSION

We introduce $A^\star$MCTS, a search algorithm that uses pre-trained value/policy networks to guide exploration and learn to make optimal decisions. We provide expected sample complexity bounds and demonstrate theoretically and experimentally that our algorithms work efficiently for reasonable models of reward distributions and noise (in value/policy network estimates) distribution. Future work includes more rigorous experimental analysis, improved search algorithms that automatically adapts to the noise level of the value/policy networks, and applications to real scenarios.

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

## 8 APPENDIX

### 8.1 PROOF OF THEOREM 1

*Proof.* Firstly, the event $\mathcal{E}$ that all states $s_d$ satisfy $|X_{s_d}| = |V_{s_d} - U_{s_d}| \leq c_d$ happens with probability $1 - \beta$, where $X_{s_d}$ are i.i.d. copies of $X_d$. This is because that $c_d$ is chosen so that $\mathbb{P}(|V_{s_d} - U_{s_d}| \leq c_d) \geq 1 - \frac{\beta}{DK^d}$ for each $s_d$. Therefore, taking union bound over $K^d$ states at depth $d$ and then over all $D$ levels ensures that the event $\mathcal{E}$ happens with probability $1 - \beta$.

$A^\star$MCTS-$V^*$ chooses iteratively the state $s_d$ with the highest UCB estimate, $U_{s_d} + c_d$ and queries the value network for the values of each of the $K$ children of $s_d$. Under event $\mathcal{E}$, the UCB estimate $U_{s_d} + c_d$ is always an over-estimate of $V_{s_d}$. Since the algorithm always take from the queue the state $s_d$ with the highest UCB estimate $U_{s_d} + c_d$ and it only stops when the queue $Q$ of possible candidates has only one leaf element, the algorithm is guaranteed to find $l^*$. Suppose otherwise, i.e., $|Q| = 1$ and $Q$ contains a leaf state $l$ where $V_l < V^*$. Then for an intermediate state $s_d$ that is an ancestor to $l^*$, $V_l < V_{s_d} = V^* \leq U_{s_d} + c_d$ under the event $\mathcal{E}$. Therefore, as $Q$ is sorted in descending order according to the UCB estimates, $s_d$ must be ahead of $l$ in $Q$, which is a contradiction.

To estimate the complexity, one needs to think about how many non-optimal internal nodes are queried using the value network. Notice that a sub-optimal internal node $s_d$ is chosen if $V^* \leq U_{s_d} + c_d = V_{s_d} + X_{s_d} + c_d$. Then it is necessary to understand the probability of the event that $\Delta_{s_d} - X_{s_d} - c_d \geq 0$, because under that event the algorithm does not need to query the value network for $s_d$. Therefore the sample complexity $N$ (i,e. the number of states for which the value network is queried) is the given by

$$N = \sum_{s_d \in \mathcal{S}, s_d \notin \mathcal{L}} K \cdot \mathbb{1}\left(\Delta_{s_d} - X_{s_d} - c_d \leq 0 \bigcap_{s_{d'} \in \mathcal{A}(s_d)} \Delta_{s_{d'}} - X_{s_{d'}} - c_{d'} \leq 0\right),$$

where $K$ is the number of children of each node, and $\mathcal{A}(s_d)$ is the set of ancestors of state $s_d$.

Since $V_{s_d} = V^*$, $\Delta_{s_d} = 0$ whenever $s_d \in \mathcal{A}(l^*)$. Moreover, because $|X_{s_d}| \leq c_d$ under event $\mathcal{E}$, it follows that the expected sample complexity is

$$\mathbb{E}[N] = KD + \sum_{s_d \in \mathcal{S}, s_d \notin \mathcal{L}, s_d \notin \mathcal{A}(l^*)} K \cdot \mathbb{P}\left(\Delta_{s_d} - X_{s_d} - c_d \leq 0 \bigcap_{s_{d'} \in \mathcal{A}(s_d)} \Delta_{s_{d'}} - X_{s_{d'}} - c_{d'} \leq 0\right)$$

$\square$

### 8.2 PROOF OF THEOREM 2

*Proof.* Firstly, the event $\mathcal{E}$ that all states $s_d$ satisfy $|V_{s_d} - U_{s_d}| \leq c_d$ and $|V_{s_d} - U_{s_d}^\pi| \leq c_d^\pi$ happens with probability $1 - \beta$, following the same union bound argument as in Theorem 1.

The correctness of $A^\star$MCTS-$\pi^*V^*$ follows largely from the proof of correctness for $A^\star$MCTS-$V^*$. The main difference in the two algorithms is in the new sampling condition for $r_i$ for $i > 2$, which is that $\ln\left(\frac{p_{r_1}}{p_{r_{i-1}}}\right) \leq 2c_{d+1}^\pi$. To show why this is relevant, let us elaborate on the case of $i = 3$. Under the event $\mathcal{E}$, $V_{r_3} \leq U_{r_3}^\pi + c_{d+1}^\pi$, where $c_{d+1}^\pi$ depends on the noise distribution $X_{d+1}^\pi$. Moreover, $V_{r_1} \geq U_{r_1}^\pi - c_{d+1}^\pi$. Hence, one should sample $r_3$ if $U_{r_2}^\pi + c_{d+1}^\pi \geq U_{r_1}^\pi - c_{d+1}^\pi$, i.e. $U_{r_1}^\pi - U_{r_2}^\pi \leq 2c_{d+1}^\pi$. This is because if $U_{r_2}^\pi + c_{d+1}^\pi < U_{r_1}^\pi - c_{d+1}^\pi$ then it is impossible for $V_{r_3} > V_{r_1}$.

Since the probabilities are given in a softmax form, the condition $U_{r_1}^\pi - U_{r_2}^\pi \leq 2c_{d+1}^\pi$ is equivalent to

$$\mathbb{1}\left(\ln\left(\frac{p_{r_1}}{p_{r_2}}\right) \leq 2c_{d+1}\right).$$

When combined with the result from Theorem 1, it gives the following sample complexity bound:

$$\sum_{s_d \notin \mathcal{L}}\left(2 + \sum_{k=2}^{K-1} \mathbb{1}\left(\ln\left(\frac{p_{r_1}}{p_{r_k}}\right) \leq 2c_{d+1}^\pi\right)\right)$$

$$\cdot \mathbb{1}\left(\Delta_{s_d} - X_{s_d} - c_d - c_{\widetilde{D}} \leq 0 \bigcap_{s_{d'} \in \mathcal{A}(s_d)} \Delta_{s_{d'}} - X_{s_{d'}} - c_{d'} - c_{\widetilde{D}} \leq 0\right).$$

This is equivalent to:

$$\sum_{s_d \notin \mathcal{L}} \left(2 + \sum_{k=2}^{K-1} \mathbb{1}\left(V_{r_1} - V_{r_k} + X_{(d+1),1}^{\pi} - X_{(d+1),k}^{\pi} \leq 2c_{d+1}^{\pi}\right)\right) \cdot$$

$$\mathbb{1}\left(\Delta_{s_d} - X_{s_d} - c_d - c_{\widetilde{D}} \leq 0 \bigcap_{s_{d'} \in \mathcal{A}(s_d)} \Delta_{s_{d'}} - X_{s_{d'}} - c_{d'} - c_{\widetilde{D}} \leq 0\right),$$

where $X_{(d+1),k}^{\pi}$ is the noise random variable associated with $r_k$ for the policy network, i.e., $U_{r_k}^{\pi} = V_{r_k} + X_{(d+1),k}^{\pi}$. The events $\mathbb{1}\left(V_{r_1} - V_{r_k} + X_{(d+1),1}^{\pi} - X_{(d+1),k}^{\pi} \leq 2c_{d+1}^{\pi}\right)$ and $\mathbb{1}\left(\Delta_{s_d} - X_{s_d} - c_d - c_{\widetilde{D}} \leq 0 \bigcap_{s_{d'} \in \mathcal{A}(s_d)} \Delta_{s_{d'}} - X_{s_{d'}} - c_{d'} - c_{\widetilde{D}} \leq 0\right)$ are independent, since $\Delta_{s_d}$ is the gap between $V^*$ and $V_{s_d}$, $V_{r_1} - V_{r_k}$ is the difference in value between two children of $s_d$, and these two quantities are uncorrelated.

Therefore our total expected sample complexity is:

$$\mathbb{E}[N] \leq \sum_{s_d \notin \mathcal{L}} \left(2 + \sum_{k=2}^{K-1} \mathbb{P}\left(V_{r_1} - V_{r_k} + X_{(d+1),1}^{\pi} - X_{(d+1),k}^{\pi} \leq 2c_{d+1}^{\pi}\right)\right) \cdot$$

$$\mathbb{P}\left(\Delta_{s_d} - X_{s_d} - c_d \leq 0 \bigcap_{s_{d'} \in \mathcal{A}(s_d)} \Delta_{s_{d'}} - X_{s_{d'}} - c_{d'} \leq 0\right),$$

$\square$

## 8.3 APPROXIMATING $V^*$

---

**Algorithm 3** $A^\star$MCTS-$\hat{V}$

---

**Require:** Value Network, additive approximation tolerance $\delta$, failure tolerance $\beta$
1: For each $d < D$, compute $c_d$ so that $\mathbb{P}(|V_{s_d} - U_{s_d}| \leq c_d) \geq 1 - \frac{\beta}{DK^d}$ for each $s_d$.
2: Calculate $\widetilde{D}$ such that $c_{\widetilde{D}} \leq \delta$.
3: $Q \leftarrow (s_0, U_{s_0} + c_0)$
4: **while** the depth of $Q.front()$ is $< \widetilde{D}$ **do**
5:     $(s, \_) = Q.pop()$
6:     **for** $r \in \mathcal{C}(s)$ **do**
7:         $d \leftarrow$ depth of $r$
8:         $Q.enqueue(r, U_r + c_d)$
9:     **end for**
10: **end while**
11: $(s, \_) \leftarrow Q.front()$
12: **Return** $U_s$ and the action at $s_0$ that leads to $s$

---

### 8.3.1 PROOF OF THEOREM 3 FOR ALGORITHM 3

*Proof.* Under event $\mathcal{E}$, the correctness for Algorithm 3, $A^\star$MCTS-$\hat{V}$ largely follows the proof in Theorem 1, with the exception that the algorithm stops when the most optimistic state in the queue $Q$ is at a special intermediate depth $\widetilde{D}$. Let $\tilde{l}^*$ be the state at depth $\widetilde{D}$ that achieves $\arg\max_{s_d \in Q} U_{s_d} + c_d$.

For every node $s_{\widetilde{D}}$ at depth $\widetilde{D}$, $Q$ contains an ancestor of $s_{\widetilde{D}}$. Under the event $\mathcal{E}$, every estimate $U_{s_d} + c_d$ is a valid upper bound of $V_{s_d}$ and the true value of $s_d$'s descendants. It follows that $U_{\widetilde{l}^*} - c_{\widetilde{D}} \leq V^* \leq U_{\widetilde{l}^*} + c_{\widetilde{D}}$. Therefore, $U_{\widetilde{l}^*}$ approximates $V^*$ up to additive error $\delta$ when $c_{\widetilde{D}} \leq \delta$.

In order to reason about the sample complexity, notice that we query the value network $U$ for a state $s_d$ if $U_{\widetilde{l}^*} < U_{s_d} + c_d \leq V_{s_d} + X_{s_d} + c_d$ with the additional requirement that $d \leq \widetilde{D}$. Therefore, it is sufficient to consider the event where $U_{\widetilde{l}^*} - V_{s_d} \leq X_{s_d} + c_d$. Since $U_{\widetilde{l}^*}$ is lower-bounded by $V^* - c_{\widetilde{D}}$, it suffices to consider the event of $V^* - V_{s_d} \leq X_{s_d} + c_d + c_{\widetilde{D}}$.

Therefore the sample complexity $N$ of states queried using the value network is upper-bounded by

$$N \leq \sum_{s_d \in \mathcal{S}, d < \widetilde{D}} K \cdot \mathbb{1}\left( \Delta_{s_d} - X_{s_d} - c_d - c_{\widetilde{D}} \leq 0 \bigcap_{s_{d'} \in \mathcal{A}(s_d)} \Delta_{s_{d'}} - X_{s_{d'}} - c_{d'} - c_{\widetilde{D}} \leq 0 \right).$$

Since $\Delta_{s_d} = 0$ whenever $s_d \in \mathcal{A}(l^*)$ and $|X_{s_d}| \leq c_d$ under event $\mathcal{E}$, it follows that the expected sample complexity is

$$\mathbb{E}[N] \leq K\widetilde{D} + \sum_{\substack{s_d \in \mathcal{S} \\ d < \widetilde{D} \\ s_d \notin \mathcal{A}(l^*)}} K \cdot \mathbb{P}\left( \Delta_{s_d} - X_{s_d} - c_d - c_{\widetilde{D}} \leq 0 \bigcap_{s_{d'} \in \mathcal{A}(s_d)} \Delta_{s_{d'}} - X_{s_{d'}} - c_{d'} - c_{\widetilde{D}} \leq 0 \right).$$

$\square$

## 8.4 CONSTANT GAP MODEL

The first example we will consider is the constant gap model, where $\Delta(l) = \eta$ for any $l \neq l^*$, and the noise variables $X_d, X_d^\pi$ are mean-0 Gaussian random variables.

### 8.4.1 USING VALUE NETWORK

**Lemma 1.** *In the Constant Gap Model, the expected sample complexity for Algorithm 1 is:*

$$\mathbb{E}[N] = KD + \sum_{d=1}^{D-1} \sum_{i=0}^{d-1} \left[ (K-1) \prod_{j=0}^{i} K\mathbb{P}(\eta - X_{(d-j)} - c_{(d-j)} \leq 0) \right].$$

*This translates to a sample complexity of $\mathbb{E}[N] \leq KD + D^2(K-1)K^c$, for some constant $c$ that satisfies $\frac{\eta}{\sigma_c} - \sqrt{c} \geq \sqrt{2\log K}$.*

*Proof.* For $A^\star$MCTS-$V^*$, Algorithm 1, Theorem 1, the sample complexity

$$\mathbb{E}[N] = KD + \sum_{s_d \in \mathcal{S}, s_d \notin \mathcal{L}, s_d \notin \{\mathcal{A}(l^*)\}} K \cdot \mathbb{P}\left( \Delta_{s_d} - X_{s_d} - c_d \leq 0 \bigcap_{s_{d'} \in \mathcal{A}(s_d)} \Delta_{s_{d'}} - X_{s_{d'}} - c_{d'} \leq 0 \right),$$

is easy to evaluate, because the individual events $\Delta_{s_d} - X_{s_d} - c_d \leq 0$ and $\Delta_{s_{d'}} - X_{s_{d'}} - c_{d'} \leq 0$ are independent. We claim that

$$\sum_{s_d \in \mathcal{S}, s_d \notin \mathcal{L}, s_d \notin \{\mathcal{A}(l^*)\}} K \cdot \mathbb{P}\left( \Delta_{s_d} - X_{s_d} - c_d \leq 0 \bigcap_{s_{d'} \in \mathcal{A}(s_d)} \Delta_{s_{d'}} - X_{s_{d'}} - c_{d'} \leq 0 \right)$$

is equivalent to

$$\sum_{d=1}^{D-1} \sum_{i=0}^{d-1} \left[ (K-1) \prod_{j=0}^{i} K\mathbb{P}(\eta - X_{(d-j)} - c_{(d-j)} \leq 0) \right]. \tag{2}$$

To see why this is the case, we can reason about the first few levels. At depth $d = 1$, we choose to expand a state with probability $\mathbb{P}(\eta - X_1 - c_1 \leq 0)$. There are $K - 1$ states $s_1$ where $s_1 \notin \mathcal{A}(l^*)$.

If we expand a particular node, the sample complexity is $K$. Therefore the expression would be $(K - 1)(K) \cdot \mathbb{P}(\eta - X_1 - c_1 \leq 0)$.

At depth $d = 2$, there are $(K - 1)(K)$ states whose depth 1 ancestor is not in $\mathcal{A}(l^*)$. For these states, we choose to expand each with probability $\mathbb{P}(\eta - X_2 - c_2 \leq 0)\mathbb{P}(\eta - X_1 - c_1 \leq 0)$. The sample complexity for each expansion is $K$, so in expectation, we query $(K - 1)(K^2)\mathbb{P}(\eta - X_2 - c_2 \leq 0)\mathbb{P}(\eta - X_1 - c_1 \leq 0)$ states. There are also $K - 1$ states who are not in $\mathcal{A}(l^*)$ but their parent in depth 1 are in $\mathcal{A}(l^*)$, and we expand them with probability $\mathbb{P}(\eta - X_2 - c_2 \leq 0)$. For these states, we query $(K - 1)(K)\mathbb{P}(\eta - X_2 - c_2 \leq 0)$ states in expectation. Continuing this reasoning through for depths $3, \ldots D - 1$ yields the final expression in Equation 2.

Therefore the expected sample complexity is:

$$\mathbb{E}[N] = KD + \sum_{d=1}^{D-1} \sum_{i=0}^{d-1} \left[ (K - 1) \prod_{j=0}^{i} K\mathbb{P}(\eta - X_{(d-j)} - c_{(d-j)} \leq 0) \right] .$$

We can rewrite the above as:

$$\mathbb{E}[N] = KD + \sum_{d=1}^{D-1} \sum_{i=0}^{d-1} \left[ (K - 1) \prod_{j=0}^{i} K\mathbb{P}(X_{(d-j)} \geq \eta - c_{(d-j)}) \right] .$$

which is equivalent to:

$$\mathbb{E}[N] = KD + \sum_{d=1}^{D-1} \sum_{i=0}^{d-1} \left[ (K - 1) \prod_{j=0}^{i} K\mathbb{P}\left( Z \geq \frac{\eta}{\sigma_{(d-j)}} - \sqrt{(d - j)} \right) \right] ,$$

where $Z$ is the standard normal distribution.

Standard tail inequalities gives us that $\mathbb{P}(Z \geq t) \leq \frac{1}{t} \frac{1}{\sqrt{2\pi}} e^{-\frac{t^2}{2}} \leq e^{-\frac{t^2}{2}}$, where the last inequality holds for $t > 1$.

We are interested in a value for $t$ such that $e^{-\frac{t^2}{2}} \leq e^{-\log K}$. This is achieved when $t = \sqrt{2 \log K}$.

For exponentially decaying standard deviation as a function of depth, i.e., $\sigma_d = \frac{1}{\alpha^d}$ for $\alpha > 1$, when $\eta \cdot \alpha^{(d-j)} - \sqrt{d - j} > t$, $\mathbb{P}\left( Z \geq \frac{\eta}{\sigma_{(d-j)}} - \sqrt{(d - j)} \right) \leq \frac{1}{K}$.

Since $\alpha$, and $K$ are constants, there exists a constant $c$ such that

$$\eta \cdot \alpha^{(d-j)} - \sqrt{(d - j)} \geq \sqrt{2 \log K}$$

whenever $d - j \geq c$.

Therefore, the sample complexity is bounded by:

$$\mathbb{E}[N] \leq KD + D^2(K - 1)K^c \tag{3}$$

which is polynomial in the depth $D$.

For polynomially decaying standard deviation as a function of depth, i.e., $\sigma_d = \frac{1}{d^\alpha}$ for $\alpha > 1$, when $\eta \cdot (d - j)^\alpha - \sqrt{d - j} > t$, $\mathbb{P}\left( Z \geq \frac{\eta}{\sigma_{(d-j)}} - \sqrt{(d - j)} \right) \leq \frac{1}{K}$.

Since $\alpha$, and $K$ are constants, there exists a constant $c'$ such that

$$\eta \cdot (d - j)^\alpha - \sqrt{(d - j)} \geq \sqrt{2 \log K}$$

whenever $d - j \geq c'$, and a similar sample complexity to Equation 3 holds. $\qquad\square$

### 8.4.2 $\delta$-APPROXIMATE VALUE ESTIMATION

The above reasoning generalizes naturally to a $\delta$-approximate value estimation using $A^\star\text{MCTS-}\hat{V}$, Algorithm 3, and the expected sample complexity is:

$$\mathbb{E}[N] = K\widetilde{D} + \sum_{d=1}^{\widetilde{D}-1} \sum_{i=0}^{d-1} \left[ (K - 1) \prod_{j=0}^{i} K\mathbb{P}(\eta - X_{(d-j)} - c_{(d-j)} - c_{\widetilde{D}} \leq 0) \right] .$$

### 8.4.3 USING VALUE AND POLICY NETWORK

**Lemma 2.** *In the Constant Gap Model, the expected sample complexity for Algorithm 2 is:*

$$\mathbb{E}[N] = \sum_{d=0}^{D-1} \left( 2 + \sum_{k=2}^{K-1} \mathbb{P}\left(\eta + X_{(d+1),1} + X_{(d+1),k} \leq 2c_{d+1}^{\pi}\right) \right)$$

$$+ \sum_{d=1}^{D-1} \sum_{i=0}^{d-1} \left[ 2 + \sum_{k=2}^{K-1} \mathbb{P}\left(\eta + X_{(d-i),1} + X_{(d-i),2} \leq 2c_{d-i}^{\pi}\right) \right] \prod_{j=0}^{i} K\mathbb{P}(\eta - X_{d-j} - c_{d-j} \leq 0)$$

*Proof.* The policy network for this constant gap problem model is not extremely useful, because for suboptimal states, the immediate children have the same optimality gap, and therefore, the probabilities given by the policy network will be fairly evenly distributed among the immediate children nodes. The policy network can give more differentiated probabilities for the immediate children of a parent state that is an ancestor of the optimal leaf.

First, the sample complexity $KD$ for the optimal path can be improved to

$$\sum_{d=0}^{D-1} \left( 2 + \sum_{k=2}^{K-1} \mathbb{P}\left(\eta + X_{(d+1),1} + X_{(d+1),k} \leq 2c_{d+1}^{\pi}\right) \right) .$$

For the ancestor leaf at level $d$, the number of child leaves that we would expand in expectation using the information from the policy network is $\left( 2 + \sum_{k=2}^{K-1} \mathbb{P}\left(\eta + X_{(d+1),1} + X_{(d+1),k} \leq 2c_{d+1}^{\pi}\right) \right)$.

The second expression can be improved to:

$$\sum_{d=1}^{D-1} \sum_{i=0}^{d-1} \left[ 2 + \sum_{k=2}^{K-1} \mathbb{P}\left(\eta + X_{(d-i),1} + X_{(d-i),2} \leq 2c_{d-i}^{\pi}\right) \right] \prod_{j=0}^{i} K\mathbb{P}(\eta - X_{d-j} - c_{d-j} \leq 0) .$$

To see why this is the case, we can reason about the first few levels. At depth $d = 1$, we choose to expand a state with probability $\mathbb{P}(\eta - X_1 - c_1 \leq 0)$. There are $K - 1$ states $s_1$ where $s_1 \notin \mathcal{A}(l^*)$, but in expectation we only consider $2 + \sum_{k=2}^{K-1} \mathbb{P}\left(\eta + X_{(1),1} + X_{(1),2} \leq 2c_1^{\pi}\right)$ of these states based on the information from the policy network. If we expand a particular state, the sample complexity is $K$, since all $K$ children nodes have the same value in this constant gap model, so the sampling criteria using probabilities from the policy network will always be met. Therefore the expression would be $\left[ 2 + \sum_{k=2}^{K-1} \mathbb{P}\left(\eta + X_{(1),1} + X_{(1),2} \leq 2c_1^{\pi}\right) \right] (K) \cdot \mathbb{P}(\eta - X_1 - c_1 \leq 0).$

At depth $d = 2$, there are in expectation $\left[ 2 + \sum_{k=2}^{K-1} \mathbb{P}\left(\eta + X_{(1),1} + X_{(1),2} \leq 2c_1^{\pi}\right) \right] (K)$ states that we would consider expanding from whose depth 1 ancestor is not in $\mathcal{A}(l^*)$. For these states, we choose to expand each with probability $\mathbb{P}(\eta - X_2 - c_2 \leq 0)\mathbb{P}(\eta - X_1 - c_1 \leq 0)$. The sample complexity for each expansion is $K$, so in expectation, we query $\left[ 2 + \sum_{k=2}^{K-1} \mathbb{P}\left(\eta + X_{(1),1} + X_{(1),2} \leq 2c_1^{\pi}\right) \right] (K^2)\mathbb{P}(\eta - X_2 - c_2 \leq 0)\mathbb{P}(\eta - X_1 - c_1 \leq 0)$ states. There are also $K - 1$ states who are not in $\mathcal{A}(l^*)$ but their parent in depth 1 are in $\mathcal{A}(l^*)$, and we expand them with probability $\mathbb{P}(\eta - X_2 - c_2 \leq 0)$. For these $K - 1$ states, in expectation, we have expanded only $\left[ 2 + \sum_{k=2}^{K-1} \mathbb{P}\left(\eta + X_{(2),1} + X_{(2),2} \leq 2c_1^{\pi}\right) \right]$ of them, therefore, we expect to query $\left[ 2 + \sum_{k=2}^{K-1} \mathbb{P}\left(\eta + X_{(1),1} + X_{(1),2} \leq 2c_1^{\pi}\right) \right] (K)\mathbb{P}(\eta - X_2 - c_2 \leq 0)$ states in expectation. Continuing this reasoning through for depths $3, \ldots D - 1$ yields the final expression:

$$\mathbb{E}[N] = \sum_{d=0}^{D-1} \left( 2 + \sum_{k=2}^{K-1} \mathbb{P}\left(\eta + X_{(d+1),1} + X_{(d+1),k} \leq 2c_{d+1}^{\pi}\right) \right)$$

$$+ \sum_{d=1}^{D-1} \sum_{i=0}^{d-1} \left[ 2 + \sum_{k=2}^{K-1} \mathbb{P}\left(\eta + X_{(d-i),1} + X_{(d-i),2} \le 2c_{d-i}^\pi\right) \right] \prod_{j=0}^{i} K \mathbb{P}(\eta - X_{d-j} - c_{d-j} \le 0)$$

$\square$

For a $\delta$-approximate value estimate, similar arguments apply to give a sample complexity of:

$$\mathbb{E}[N] = \sum_{d=0}^{d'-1} \left( 2 + \sum_{k=2}^{K-1} \mathbb{P}\left(\eta + X_{(d+1),1} + X_{(d+1),k} \le 2c_{d+1}^\pi\right) \right)$$

$$+ \sum_{d=1}^{d'-1} \sum_{i=0}^{d-1} \left[ 2 + \sum_{k=2}^{K-1} \mathbb{P}\left(\eta + X_{(d-i),1} + X_{(d-i),2} \le 2c_{d-i}^\pi\right) \right] \prod_{j=0}^{i} K \mathbb{P}(\eta - X_{d-j} - c_{d-j} - c_{\widetilde{D}} \le 0).$$

### 8.5 Generative Model

The model generates the value function of the nodes in the Markov decision tree in a recursive way. This model first chooses the optimal value $V_{s_0} = V^*$ at the root $s_0$. Given the value function of a parent node $p$, the value functions for the children $r_1, \ldots, r_K$ are generated as follows: one of the children (without loss of generality, say $r_1$) has the same value function as the parent $V_p$; each of the remaining children $r_i$ has a value function $V_p - Y$ with $Y$ sampled independently from $\sim U[0, \eta]$, the uniform distribution over the interval $[0, \eta]$. At level $d$, there are exactly $\binom{d}{t}(K-1)^t$ states with value function distributed according to $V^* - (Y_1 + \ldots + Y_t)$. When $t$ is non-negligible, $Y_1 + \ldots + Y_t$ concentrates at constant value $t\eta/2$ by the concentration of measure phenomenon.

**Using value networks.** Let us consider the expected complexity of Algorithm 1 for this model. In the algorithms, $c_d = \widetilde{C}\sqrt{d}\sigma_d$ for some constant $\widetilde{C}$. Following the estimate in Theorem 1, the overall complexity can be bounded up to an extra $KD$ term by

$$\sum_{d=1}^{D} \sum_{t=0}^{d} \binom{d}{t}(K-1)^t \mathbb{P}\left(X_d \ge \frac{\eta t}{2} - c_d\right) = \sum_{d=1}^{D} \sum_{t=0}^{d} \binom{d}{t}(K-1)^t \mathbb{P}\left(Z \ge \frac{\eta t}{2\sigma_d} - \widetilde{C}\sqrt{d}\right)$$

where $Z$ is a standard normal distribution. The goal is to bound this by a quantity polynomial (rather than exponential) in $D$ when $\eta$, $K$, and $\alpha$ are considered fixed. To proceed, we introduce $T(d) = \frac{2(\sqrt{2\log K}+1)\sqrt{d}\sigma_d}{\eta}$. For any $t \ge T(d)$,

$$\mathbb{P}\left(Z \ge \frac{\eta t}{2\sigma_d} - \widetilde{C}\sqrt{d}\right) \lesssim e^{-\log Kd} = K^{-d}.$$

Therefore, for a fixed $d$,

$$\sum_{t=T(d)}^{d} \binom{d}{t}(K-1)^t \mathbb{P}\left(Z \ge \frac{\eta t}{2\sigma_d} - \widetilde{C}\sqrt{d}\right) \le \sum_{t=T(d)}^{d} \binom{d}{t}(K-1)^t K^{-d} \le \sum_{t=0}^{d} \binom{d}{t}(K-1)^t K^{-d} = 1.$$

Therefore, up to a term linear in $D$, the sample complexity can be bounded by

$$\sum_{d=1}^{D} \sum_{t=1}^{T(d)} \binom{d}{t}(K-1)^t \mathbb{P}\left(Z \ge \frac{\eta t}{2\sigma_d} - \widetilde{C}\sqrt{d}\right). \tag{4}$$

Notice first that $\mathbb{P}(Z \ge \frac{\eta t}{2\sigma_d} - \sqrt{d})$ is always bounded by 1. Second, since in both the exponential and polynomial noise models $\sigma_d$ decays faster than $1/\sqrt{d}$ asymptotically, $T(d) = \frac{2(\sqrt{2\log K}+1)\sqrt{d}\sigma_d}{\eta}$ is uniformly bounded by a constant $C(K, \eta, \alpha)$ due to the exponential decay in $d$. Therefore, the term (4) can be bounded by $(DK)^{C(K,\eta,\alpha)}$, which is polynomial (rather than exponential) in the depth $D$.

**Using value and policy networks.** Let us consider now the expected complexity of Algorithm 2. Following the argument of the constant gap model, it can be bounded by

$$\sum_{d=1}^{D} \sum_{t=1}^{T(d)} \binom{d}{t}(K-1)^{t-1} \left[ 2 + \sum_{k=2}^{K-1} \mathbb{P}\left(\eta + X_{t,1}^\pi + X_{t,2}^\pi \le 2c_t^\pi\right) \right] \mathbb{P}\left(Z \ge \frac{\eta t}{2\sigma_d} - \widetilde{C}\sqrt{d}\right),$$

which is also polynomial in $D$ as it improves over the complexity of using just the value network.

**Approximating $V^*$ using value and policy networks.** For a $\delta$-approximate value estimate using a value network, one can stop early at level $\widetilde{D}$ instead and the complexity analysis gives a bound

$$\sum_{d=1}^{\widetilde{D}} \sum_{t=1}^{T(d)} \binom{d}{t} (K-1)^t \mathbb{P}\left(Z \geq \frac{\eta t}{2\sigma_d} - \widetilde{C}\sqrt{d}\right) = O((\widetilde{D}K)^{C(K,\eta,\alpha)}).$$

For a $\delta$-approximate value estimate using both policy and value networks, one can stop early at level $\widetilde{D}$ instead and the complexity analysis gives a bound

$$\sum_{d=1}^{\widetilde{D}} \sum_{t=1}^{T(d)} \binom{d}{t} (K-1)^{t-1} \left[2 + \sum_{k=2}^{K-1} \mathbb{P}\left(\eta + X_{t,1}^{\pi} + X_{t,2}^{\pi} \leq 2c_t^{\pi}\right)\right] \mathbb{P}\left(Z \geq \frac{\eta t}{2\sigma_d} - \widetilde{C}\sqrt{d}\right).$$

