# OpenReview forum: "A⋆MCTS: SEARCH WITH THEORETICAL GUARANTEE USING POLICY AND VALUE FUNCTIONS"
_ICLR.cc/2020/Conference — Reject_

### Official Review · AnonReviewer2 · 2019-10-23
**Official Blind Review #2**

**Rating:** 1

**Review:**

The paper presents a novel search algorithm that uses the policy and value predictors to guide search and provides theoretical guarantee on the sample complexity. The aim is to estimate the optimal value of an initial state as well as the one-step optimal action to take.
The algorithm uses a priority queue to store all states being visited so far and picks the most optimistic one to expand, according to an upper confidence bound heuristic function. The algorithm assumes access to pre-trained value and policy networks and it uses calls to these networks to prioritize the next state to be explored.

The authors consider a very restrictive setting:
- Finite horizon Markov decision tree:  no backtrack
- No intermediate reward and only reward at the end of the episode.
- Deterministic transition
- Importantly, access to value network that gives noisy estimates of the optimal value function
- The noise model is additive and i.i.d and satisfies a concentration inequality

All this assumption makes the setting very simple and unrealistic. Moreover, I think we can frame the problem into bandit problem and solve it easily with sample complexity independent of the horizon D.
In fact, given an initial state s, we consider the K possible actions  a_1, a_2, …, a_K that lead deterministically to next states (r_1, r_2, …, r_K). As the intermediate reward is zero, the state-action value of (s, a_k) is equal to V_{r_k}. As we have noisy estimates of V_{r_k} and we know precisely the noise model, we can run UCB-like algorithm for multi-armed bandit where each arm corresponds to action a_k and expected reward correspond to V_{r_k}. This determines the optimal action in constant time with respect to the horizon.

**Experience Assessment:**

I have published one or two papers in this area.

**Review Assessment: Checking Correctness Of Derivations And Theory:**

I assessed the sensibility of the derivations and theory.

**Review Assessment: Checking Correctness Of Experiments:**

I assessed the sensibility of the experiments.

**Review Assessment: Thoroughness In Paper Reading:**

I made a quick assessment of this paper.

---

> ### Author Response · Authors · 2019-11-07
> **thanks for your comments**
>
> We think that this reviewer misunderstood our problem setting. The value network is a pre-trained and *deterministic* function that outputs the same value estimate for a particular node, no matter how many times it is called. This is standard in applications like AlphaZero. In our formulation, the noise random variable $X_d$ was instantiated once at each particular node and is fixed afterwards. Therefore, one cannot call the value network repeatedly on the same node and average over the results to achieve the true value. Instead, one must expand further down to children nodes, because the value network estimates for the children nodes are less noisy.
>
> The finite horizon Markov Decision tree is standard in applications, in our opinion. We are not aware of any study on infinite horizon Markov Decision trees.
>
> We are not sure what you mean by “no backtracking”. If you mean the averaging operation following the parents from a new node in the regular MCTS, our algorithm also has it, by always replacing the old value of a node with the new one from more accurate child estimates.
>
> The transition and reward assumptions are valid for a lot of applications, ie games. These assumptions are also a good starting point on which to develop our results because they make the problem simpler while preserving the main challenges that we want to tackle. The value network that we assume in our paper is increasingly becoming an integral part of deep reinforcement learning.

---

### Official Review · AnonReviewer1 · 2019-10-29
**Official Blind Review #1**

**Rating:** 3

**Review:**

This paper proposes A*MCTS, which combines A* and MCTS with policy and value networks to prioritize the next state to be explored. It further establishes the sample complexity to determine optimal actions. Experimental results validate the theoretical analysis and demonstrate the effectiveness of A*MCTS over benchmark MCTS algorithms with value and policy networks.

Pros:
This paper presents the first study of tree search for optimal actions in the presence of pretrained value and policy networks. And it combines A* search with MCTS to improve the performance over the traditional MCTS approaches based on UCT or PUCT tree policies. Experimental results show that the proposed algorithm outperform the MCTS algorithms.

Cons:
However, there are several issues that should be addressed including the presentation of the paper:
•	The algorithm seeks to combine A* search with MCTS (combined with policy and value networks), and is shown to outperform the baseline MCTS method. However, it does not clearly explain the key insights of why it could perform better. For example, what kind of additional benefit will it bring when integrating the priority queue into the MCTS algorithms? How could it improve over the traditional tree policy (e.g., UCT) for the selection step in MCTS? These discussions are critical to understand the merit of the proposed algorithms. In addition, more experimental analysis should also be presented to support why such a combination is the key contribution to the performance gain.
•	Many design choices for the algorithms are not clearly explained. For example, in line 8 of Algorithm 2, why only the top 3 child nodes are added to the queue?
•	The complexity bound in Theorem 1 is hard to understand. It does not give the explicit relations of the sample complexity with respect to different quantities in the algorithms. In particular, the probability in the second term of Theorem 1 is hard to parse. The authors need to give more discussion and explanation about it. This is also the case for Theorems 2-4. The authors give some concrete examples in Section 6.2 for these bounds. However, it would be better to have some discussion earlier right after these theorems are presented.
•	The experimental results are carried out under the very simplified settings for both the proposed algorithm and the baseline MCTS. In fact, it is performed under the exact assumption where the theoretical analysis is done for the A*MCTS. This may bring some advantage for the proposed algorithm. It is not clear whether such assumptions hold for practical problems. More convincing experimental comparison should be done under real environment such as Atari games (by using the simulator as the environment model as shown in [Guo et al 2014] “Deep learning for real-time atari game play using offline monte-carlo tree search planning”).

Other comments:
•	It is assumed that the noise of value and policy network is zero at the leaf node. In practice, this is not true because even at the leaf node the value could still be estimated by an inaccurate value network (e.g., AlphaGo or AlphaZero). How would this affect the results?
•	In fact, the proof of the theorems could be moved to appendices.
•	In the first paragraph of Section 6.2, there is a typo: V*=V_{l*}=\eta should be V*-V_{l*}=\eta ?

**Experience Assessment:**

I have published one or two papers in this area.

**Review Assessment: Checking Correctness Of Derivations And Theory:**

I assessed the sensibility of the derivations and theory.

**Review Assessment: Checking Correctness Of Experiments:**

I assessed the sensibility of the experiments.

**Review Assessment: Thoroughness In Paper Reading:**

I read the paper thoroughly.

---

> ### Author Response · Authors · 2019-11-07
> **thanks for your comments**
>
> We address the concerns in the "Cons" section in order.
>
> 1. [Our scheme vs MCTS] We hypothesize that the averaging in the back-propagation step in MCTS is ineffective, since the goal is to estimate the value of the optimal policy, any averaging will inevitably bring down this value. Note that back propagation in classical value iteration, for example, takes the max, rather than the average of the different Q values. Our technique does not use the averaging scheme in MCTS.
>
> However, we are hesitant to put such a big emphasis on this intuition because there is little theoretical understanding behind MCTS and UCT applied on the entire search tree. It could be possible that the averaging could alleviate some issues in MCTS (e.g., when the value function of a node over-estimates). While we leave this to future work, the main goal of our work is to provide a tree search algorithm with provable guarantees that also works well in practice, and we think that our contribution is valuable to the ICLR community.
>
> 2. Line 8 of Algorithm 2 has an “or” condition which is crucial, we do not just add the top 3 child nodes. Depending on the probabilities given by the policy network, we may add all the children nodes to the queue. We add the top 2 (ordered by the probabilities given by the policy network) to the queue by default. For the k-th child node, where k > 2, we add this child if there is a small gap between the probability of the k-1-th child and the probability of the top child, as given by the policy network, where the threshold for the gap is given by the noise model. Intuitively, we are saying that if the gap is sufficiently big, even after accounting for the noise in the policy network, there is no way that this k-th child is part of the optimal policy, so we do not have to add it to the queue. The proof for Theorem 2 establishes in Section 4.1 why our condition is sufficient. We can add more explanation for this.
>
> 3. The complexity bound in Theorem 1 is actually based on one very simple observation, which we explain in the proof of Theorem 1. A sub-optimal internal node s is chosen if V^* <= U_s + c_d = V_s + X_s + c_d. U_s is the value network estimate for state s, c_d is the upper bound on the possible error for the value estimate. So we end up choosing a sub-optimal node if the value network estimate plus the upper bound on the possible error (optimism) is higher than the true optimal value. Otherwise we will not choose to expand that node, since we are sure that it will not be the optimal node. U_s is equal to V_s (true value of node s) plus X_s, which is the noise random variable. V^* - V_s is the expression for the gap of state s, therefore we end up with the expression in Theorem 1. Notice that we should also account for the ancestors of s, if we are lucky enough to be able to rule out one of the ancestors of s, we would never expand beyond that ancestor to reach s, and s would not be in our priority queue. We can give more explanation (and/or provide graphical illustration) on this in the manuscript, and we can introduce the concrete examples earlier.
>
> 4. [On experiments] We give a general analysis for A*MCTS in our main theorems that can be applied to a broad range of problems and models. In this paper, our main focus is on developing a principled search algorithm and providing provable theoretical guarantees, which is completely lacking in this space. The selection of models is not an emphasis of our paper, they mainly offer a sanity check that our algorithm also performs well in practice. That being said, we will definitely perform more elaborate experiments in the real situations (e.g., Atari or even self-play in AlphaZero) in the future work.
>
> 5. We can handle the case where there is error even at the leaf node. If there is error even at the leaf node, then our scheme would produce an approximation of the optimal value and an approximately optimal policy, this would be exactly similar to the results in Section 5. In Section 5, we consider the use case where we are willing to tolerate an approximately optimal policy and an approximately optimal value estimate, this basically means that we are willing to stop at a depth $\tilde D$, where the noise at level $\tilde D$ is smaller than the approximation tolerance. When there is error even at the leaf nodes, then we will find an approximately optimal policy and value estimate, where the approximation depends on the error at the leaf nodes. It also makes sense, intuitively, that getting an approximation is the best that one can hope for when there is error even at the leaf nodes.

---

### Official Review · AnonReviewer4 · 2019-11-02
**Official Blind Review #4**

**Rating:** 6

**Review:**

This paper presents the search algorithm A*MCTS to find the optimal policies for problems in Reinforcement Learning. In particular, A*MCTS combines the A* and MCTS algorithms to use the pre-trained value networks for facilitating the exploration and making optimal decisions. A*MCTS refers the value network as a black box and builds a statistical model for the prediction accuracies, which provides theoretical guarantees for the sample complexity. The experiments verify the effectiveness of the proposed A*MCTS.

In summary, I think the proposed A*MCTS algorithm is promising to push the frontier of studies of the tree search for optimal actions in RL. But the experiments should be improved to illustrate the reasons for the hyper-param setting. For example, in Sec. 6.2, the authors should give some explanations on why the depth of the tree is set as 10 and the number of children per state is set as 5.



**Experience Assessment:**

I have read many papers in this area.

**Review Assessment: Checking Correctness Of Derivations And Theory:**

I assessed the sensibility of the derivations and theory.

**Review Assessment: Checking Correctness Of Experiments:**

I carefully checked the experiments.

**Review Assessment: Thoroughness In Paper Reading:**

I read the paper thoroughly.

---

> ### Author Response · Authors · 2019-11-07
> **thanks for your comments**
>
> Our main focus is on developing a principled prioritized search algorithm with a deep learning component (e.g., Monte Carlo Tree Search algorithm with policy/value network) with provable theoretical guarantees, which is currently lacking in this space. The selection of models is not an emphasis of our paper, they mainly offer a sanity check that our algorithm also performs well in practice. We set the parameters for the tree so that we would have a reasonable model that could offer a valid sanity check.

---

### Author Response · Authors · 2019-11-14
**rebuttal**

Summary of revisions to the manuscript

1) moved the proofs to the appendix
2) added an intuition paragraph to each section
3) added illustrations to further explain the main techniques and the models
4) added a notation table to summarize notations used
5) added explanations in the main contributions section to clarify the deterministic nature of the noisy value and policy networks and explain at a high level our algorithmic ideas.

---

### Decision · Program_Chairs · 2019-12-19

**Decision:**

Reject

**Comment:**

This paper proposed an extension of the Monte Carlos Tree Search to find the optimal policy. The method combines A* and MCTS algorithms to prioritize the state to be explored. Compare with traditional MCTS based on UCT, A* MCTS seem to perform better.

One concern of the reviewers is the paper's presentation, which is hard to follow. The second concern is the strong restriction of assumption, which make the setting too simple and unrealistic. The rebuttal did not fully address these problems.

This paper needs further polish to meet the standard of ICLR.